Online shopping consumer perception analysis and future network security service technology using logistic regression model

Lu Feng lufeng@hnp.edu.cn
Department of Network Security, Henan Police College , Zhengzhou , China
Coelho Paulo
Electronic publication date: 2024 Jan 15
Publication date: 2024
Volume: 10
Electronic Location ID: e1777
Received 2023 Sep 7; Accepted 2023 Dec 5
Copyright: ©2024 Lu
Copyright year: 2024
Copyright holder: Lu
License: This is an open access article distributed under the terms of the Creative Commons Attribution License, which permits unrestricted use, distribution, reproduction and adaptation in any medium and for any purpose provided that it is properly attributed. For attribution, the original author(s), title, publication source (PeerJ Computer Science) and either DOI or URL of the article must be cited.
License URL: https://creativecommons.org/licenses/by/4.0/

Keywords: Online shopping, Consumer perception, Network security, Logistic regression model, Stimuli-organism-response model

Funding: The author received no funding for this work.

==============================
In order to understand consumer perception, reduce risks in online shopping, and maintain online security, this study employs data envelopment analysis (DEA) to confirm the relationship between evaluation and stimuli. It establishes a model of stimuli-organism response and uses regression analysis to explore the relationships among negative online shopping evaluations, consumer perception of risk, and consumer behavior. This study employs attribution theory to analyze the impact of evaluations on consumer behavior and assesses the role of perceived risk as a mediator. The independent variable is negative comments, the dependent variable is consumer behavior, and logistic regression is used to empirically analyze the factors influencing online shopping security. The results indicate a positive correlation between the number of negative comments and consumers’ delayed purchase behavior, with a correlation coefficient of 41%. The intensity of negative comments significantly impacts consumers’ refusal to make a purchase, with a correlation coefficient of 38%. The length of negative comments substantially influences consumers’ opposition to purchasing, also with a correlation coefficient of 38%. There is a close relationship between perceived risk and consumers’ delayed shopping behavior and the number of negative comments, with 41% and 4% correlation coefficients, respectively. Perceived risk has a relatively smaller impact on consumers’ opposition to purchase behavior, with a correlation coefficient of 27%. The length, intensity, and number of negative comments are correlated with consumers’ opposition, refusal, and delayed consumption, negatively affecting consumer intent. Additionally, negative comments are related to perceived risk and consumer behavior. Perceived risk causally influences consumer behavior, while the convenience of shopping has a relatively minor impact on online shopping security. Factors like delivery speed, buyer reviews, brand, price, and consumer perception are significantly related to online shopping security. Consumer perception has the most significant impact on online shopping security, balancing secure and fast consumption under the guarantee of user experience. Strengthening consumer perception enhances consumers’ ability to process risk information, helping them better identify risks and avoid using hazardous network software, tools, or technologies, thereby reducing potential online security risks.

Introduction

With the progression of the economy and advancements in Internet technology, the practice of online shopping has gradually evolved into a prominent method of consumer purchasing (Ma et al., 2022). The expanding Internet user base and the continual growth of its scale have contributed to the expansion of China’s online shopping market, establishing it as the world’s largest e-commerce market (Song, 2022). In a study by Tran (2020), associations were uncovered between factors such as product risk, financial risk, security risk, privacy risk, perceived satisfaction, and consumers’ inclination to engage in online purchases. Furthermore, Nguyen & Khoa (2019), in their examination of online sales in Vietnam, determined that perceived psychological benefits exerted the most substantial influence on the trust levels of Vietnamese customers participating in online transactions. In the e-commerce domain, both the perceived psychological benefits and the establishment of online trust significantly influence customers’ willingness to divulge their personal information. This, in turn, bears a distinct and notable consequence on the overall aspect of consumer information security. Concerns related to online trust and Internet security often lead people to have reservations about online shopping, making them more inclined to engage in non-shopping activities on the Internet. Zaidan & Raju (2021) pointed out that attitudes, perceived trust, and perceived security directly and significantly impact the intention to make online purchases. Among these factors, customers’ attitudes toward online shopping are considered the most crucial predictor of the willingness to shop online. Furthermore, consumer perception plays a critical role in internet security issues. Secure and reliable websites encourage an increase in online shopping, and a safe online environment affects consumers’ attitudes and decisions regarding online shopping. Consumer decisions, in turn, impact internet security issues in online shopping. Factors such as the choice of shopping channels or transactions with sellers can either increase or decrease online security risks. This interplay underscores the interrelationship between consumer perception and internet security. Therefore, studying consumer perception is essential to ensure online shopping security and enhance consumers’ willingness to shop online.

Bilal et al. (2022) established a positive correlation between consumers’ emotional attitudes and their intentions to make online purchases. Their study further elucidated the significant moderating role of social support in shaping online purchase intentions, affirming the impact of consumers’ perceptions on such intentions. In parallel research, Zia et al. (2022) identified a trend where, as the economy continues to develop, consumers exhibit greater maturity and rationality in their decision-making processes. Furthermore, the enhancement of customer service skills emerged as a key strategy to enhance consumer well-being and overall satisfaction, subsequently fostering increased trust in online platforms and merchants. These findings underscore the compelling necessity of augmenting consumers’ perception of the online shopping experience. The influence of consumers’ price perception on their online purchasing decisions is of paramount significance (Lestari, Sabur & Maulidiah, 2022). García-Salirrosas et al. (2022) conducted a survey using a questionnaire to gather data and subsequently constructed a structural equation model. This model designated purchase intention as the dependent variable, while trust, perceived value, and satisfaction were identified as determinants of purchase intention. The results revealed that trust and satisfaction had a direct and positive impact on value perception and online purchase intention. Similarly, Salem & Alanadoly (2022) observed that perceived price and the online shopping experience played pivotal roles in enhancing the online purchasing behavior of fashion products. Perceived price could mediate the relationship between opinion seeking, online shopping experience, and online purchasing behavior. Moreover, adjustments to perceived prices could substantially improve consumers’ online purchasing behavior. Apasrawirote & Yawised (2022) unveiled a noteworthy positive impact on customer perception and the marketing mix. Furthermore, content and influencer marketing positively correlated with consumer attitudes. In addition, the perceived value and attitude of consumers were identified as influential factors affecting their actions and behaviors. These findings underscore the profound influence of consumer perception on their behavioral intentions. Notably, a discernible connection between perceived value and shopping behavior intention was established, with the improvement of consumers’ perceived value serving as a catalyst for stimulating consumption (Panarello & Gatto, 2022). Abu-AlSondos et al. (2023) delineated a multitude of factors that impact online security in the realm of e-commerce. These factors encompass diverse elements, including website data protection measures, payment security, consumer perceived trust, shopping platform reputation, user account security, logistics, and delivery safety. Collectively, these factors exert a significant influence on consumers’ decision-making and behaviors in the domain of online shopping, thereby engendering direct or indirect ramifications for online security. The emergence of online security issues may encompass concerns such as data breaches, identity theft, fraudulent transactions, and deceptive advertising. Consequently, both online shopping platforms and consumers are urged to accord due importance to these factors to ensure the security and credibility of the online shopping environment. Wai et al. (2019) identified a degree of risk associated with consumers’ purchasing behavior. It is imperative to undertake ongoing research on consumer perception to effectively mitigate the potential risks associated with consumer perception, thereby averting any adverse consequences that may arise from misguided decisions, which could lead to a loss of consumer interest. Ventre & Kolbe (2020) ascertained that within the context of online shopping, consumers’ perceptions of online reviews exerted an influence on their trust in online shopping merchants and their propensity to engage in online purchases. Trust exhibited an inverse relationship with Perceived Risk (PR) while simultaneously yielding a positive impact on online purchase intentions. Risk perception is a pivotal determinant shaping how individuals evaluate risk, arrive at decisions, and exhibit corresponding behaviors. Li et al. (2020) discerned a notable inverse relationship between risk perception and consumers’ purchasing conduct, with this association particularly pronounced in the domain of online shopping. Likewise, Ngarmwongnoi et al. (2020) documented that consumers actively sought both positive and negative reviews. They observed that the presence of negative reviews had the capacity to impact consumers’ risk perception and subsequently induce alterations in their shopping behaviors.

In order to enhance online shopping security and mitigate the risks associated with online consumer transactions, it is imperative to investigate consumer perception in the context of online shopping. This study begins by utilizing the data envelopment analysis (DEA) model to substantiate the relationship between evaluation and stimulus. Subsequently, it establishes the classic Stimulus-Organism-Response (SOR) model, incorporating negative online reviews, consumer perceived risk, and consumer behavior as research variables, and conducts regression analysis to examine the relationships among these variables. Attribution theory is employed to attribute the impact of evaluations on consumer behavior. Within this framework, consumer-perceived risk serves as the mediating variable, negative reviews as the independent variable, and consumer behavior as the dependent variable. Finally, a logistic regression model is employed to empirically analyze factors that may influence online shopping security. This study’s contributions and novelty are multifaceted: 1. by utilizing the DEA model, this research elucidates the relationship between evaluation and stimulus in online shopping, thus providing an initial foundation for the study of perceived risk. 2. It establishes the classic SOR model, encompassing negative reviews, consumer perceived risk, and consumer behavior as research variables, offering a fresh perspective on comprehending consumer behavior in online shopping. 3. Through the application of attribution theory, this study elucidates the relationship between evaluation effects and consumer behavior, unveiling the mediating role of consumer perceived risk, thereby deepening the understanding of perceived risk. 4. This research integrates the SOR model with the Logistic Regression model to confirm the impact of negative reviews on consumer perceived risk and the influence of perceived risk on online shopping security. This integration provides a new theoretical framework and approach for safeguarding online shopping security, emphasizing the significance of perceived risk in online shopping safety and how enhancing consumer perception can mitigate online shopping risks.

Section 1 is the introduction. It provides the research’s background, highlights the impact of consumer perception on online shopping security and consumer behavior, and underscores the research’s significance, objectives, research methods, and contributions. This section elucidates the importance of exploring online consumer perception. By utilizing the DEA model, the classic SOR model, and the logistic regression model, it analyzes the influence of negative reviews on consumer perceived risk and the impact of perceived risk on online shopping security. This innovative study makes a valuable contribution to enhancing online shopping security, offering new theories and methods while emphasizing the critical role of perceived risk in this process.

Section 2 is a literature review. It presents common models used in researching consumer perception and their research status in domestic and international contexts. This section summarizes the significance of the SOR model in analyzing consumer behavior, such as perception, satisfaction, trust, and purchase intention, emphasizing the model’s applicability in various contexts. The section also identifies current research gaps and underscores the value of this study.

Section 3 outlines the research theory and methodology. It introduces the construction process of consumer perception in conjunction with Attribution Theory, the DEA model, consumer perception models, and Logistic Regression models. The section explicates the variables and equations of the models.

Section 4 delves into experimental design, results analysis, and discussions. It details the dataset, experimental environment, experimental results, and comparisons with prior research. The section confirms the impact of negative reviews on consumer perceived risk and the influence of perceived risk on online shopping security, offering new perspectives on safeguarding online consumer behavior and online shopping security.

Section 5 is the conclusion, summarizing the study’s findings, highlighting its limitations, and providing prospects for future research.

Literature Review

Various models and methods are employed in researching consumer perception, including logistic regression, factor analysis, partial least squares (PLS), SOR model, and DEA. Rahman & Nguyen-Viet (2023) adopted a survey approach to study the relationship between consumer perception and green brands, utilizing PLS for analysis. The research revealed that advertising acceptance and the green brand image positively influenced trust, which, in turn, positively affected purchase intent. Deceptiveness and transparency had a weaker impact on trust. Safeer & Liu (2023) indicated that combining PLS and survey methods facilitated the analysis of the relationship between perceived authenticity and product brands, contributing to enhancing brand authenticity. This, in turn, promoted brand trust, positive word-of-mouth, and brand loyalty. Since product brand and online security are closely related, renowned brands can enhance consumer trust in online security, thus stimulating purchasing behavior. Kim & Park (2019) employed the SOR model to investigate the impact of airport self-service features on consumers. Perceived value and consumer satisfaction significantly influenced consumer behavior, underscoring the study’s significance in enhancing airport services. Shi et al. (2023) contended that establishing a SOR model for live-streamed short videos helped assess user willingness to share marketing information on short video platforms. Their findings highlighted the substantial influence of information quality on user willingness to share marketing information. Mim, Jai & Lee (2022) utilized the Stimuli-Organism-Response model to examine the effects of sustainable positioning and high or low transparency from reliable sources on brand attachment, trust, and recognition. This research demonstrated the positive role of brand identification in clothing purchases. Lee & Min (2021), in their study of the influence of online travel agency information quality on consumer trust and intention to continue using, effectively employed the SOR model to elucidate the relationship among online travel agency information, consumer trust, and intent to continue using. Their research highlighted the positive impact of information quality factors such as accuracy, timeliness, and usefulness on consumer trust and intention to continue using. Mkedder, Bakir & Lachachi (2021) investigated local dairy product purchase intent using the SOR model, with perceived quality, perceived price, customer satisfaction, brand image, and customer purchase intent as research variables. They confirmed that perceived quality, customer satisfaction, and brand image positively impacted consumer purchase intent, with customer satisfaction and brand image acting as mediating factors. Perumal, Ali & Shaarih (2021) noted the significant mediating role of airline image in touch, taste, and repurchase intent in terms of sensory impact and consumer repurchase relations. The fairness of perceived pricing could alleviate the relationship between airline image and repurchase intent. Based on the SOR model, Ni et al. (2020) affirmed the relationship between product information recommended by friends and consumer purchase intent, contributing to a better understanding of consumer behavior for online retailers. Yen (2023) utilized the SOR model to illustrate the relationship between platform channels and consumer perception. The study revealed that platform channels influenced consumer intentions and trust in food delivery platforms by affecting perceived usefulness, perceived pleasure, and perceived price. Consumer trust in the platform is usually closely associated with online shopping security. When consumers trust a shopping platform, they are more likely to believe that their transactions are secure. This trust may be built on factors such as platform reputation, history, security measures, and user ratings. Conversely, if consumers do not trust the shopping platform, they may worry about their personal and financial information safety. Therefore, increasing trust in the platform can help enhance online security, as highly trusted platforms typically implement more measures to protect consumer information and reduce online risks. In the logistic regression model research, Sucipto, Yusuf & Mulyati (2022) delved into the performance of sequence models like logistic regression and long short-term memory (LSTM) in analyzing consumer sentiments. Through empirical studies on a review dataset, they found that LSTM excelled in capturing sequence features, showing advantages over traditional logistic regression. However, in certain contexts, logistic regression also demonstrated its unique superiority. The findings provided a robust reference for choosing an appropriate sequence model. Chen et al. (2022), focusing primarily on logistic regression, conducted a comprehensive analysis of negative reviews, perceived risks, and consumer behavior. They discovered that logistic regression effectively explained and predicted consumer behavior and performed exceptionally well in handling sequence data. The findings empirically supported the enhancement of consumer perception analysis using logistic regression and offered practical approaches to improving online shopping security.

In summary, prior research has primarily focused on the relationship between consumer perception and factors such as brands, products, services, and advertising, with relatively less attention given to the relationship between consumer perception and online security. However, online security plays an increasingly crucial role in consumers’ shopping experiences and decision-making in today’s digital era. Based on the above content, this study considers exploring more advanced sequence models on the foundation of logistic regression. This exploration aims to better adapt to the demands of handling sequence features such as negative reviews, contributing to an enhancement in the depth and breadth of the research. It provides a more accurate and reliable foundation for formulating online security strategies. Therefore, it is necessary to address this research gap and delve deeper into the mutual influences between consumer perception and online security to understand and address issues of security and trust in the online shopping process. This will contribute to formulating more effective online security strategies, enhancing consumers’ online shopping experiences, and promoting the sustainable development of e-commerce. In addition, prior research has largely relied on traditional methods such as the SOR model, surveys, and PLS without incorporating other models or datasets, making the research subject to a higher degree of subjectivity.

Research theory and establishment of consumer perception model

Consumer perception and attribution theory

Zaithaml first proposed the theory of consumer perception in 1988. It refers to a subjective evaluation of the utility of a product or service by consumers after perceiving the benefits of the product or service and removing the cost of acquiring the product or service (Ma, Ko & Lee, 2019; Izogo, Jayawardhena & Karjaluoto, 2023). The comprehensive meaning and definition of perceived value are shown in Fig. 1.

Consumer purchasing behavior, as a part of the information processing procedure, can be segmented into three distinct stages: disclosure, attention, and comprehension. The specific process of consumer information processing is shown in Fig. 2.

Consumer perception is a combination of consumer perception and sensation. Consumer-perceived psychological behavior is a prerequisite for other consumer psychological behaviors, which can have a certain degree of influence on consumers’ purchase intention, purchase decision, and purchase behavior (Lamonaca et al., 2022; Lee & Lin, 2022). Consumer perception risk pertains to the presence of potential uncertainties that may arise during the course of consumer decision-making in the shopping process. These elements of uncertainty, which consumers are capable of discerning, are denoted as “PRs”. The consumer purchasing pattern and the impact pattern of negative reviews are shown in Fig. 3.

Attribution theory, rooted in the examination of human behavior within the realm of social psychology, investigates the causes underlying specific behaviors. This theory posits that individuals tend to attribute the occurrence of an event to internal and external factors. In the context of consumers’ perceptions of online shopping, the application of attribution theory can dissect the manner and impetus behind a particular consumer behavior.

DEA model design

DEA is a well-established tool within the field of economic research, serving as a pivotal instrument for extracting substantial managerial insights. This method, initially formulated by Charnes and Cooper in 1978, encompasses several notable models. Among these models, the Charnes-Cooper-Rhodes model is primarily applied to the estimation of frontiers, while the Banker-Charnes-Cooper model focuses on assessing variable scale benefits (Charnes, Cooper & Rhodes, 1978) .

Suppose s is the decision unit, and Xns the input vector of the decision unit with dimension n.Ysm is the output vector produced by the decision unit with dimension m, and Z is the set of all possible outputs of Y when the input is X. Then, the effectiveness Hs of the decision unit can be calculated according to Eq. (1). (1) Hs=UZYsmVZXsn.

In Eq. (1), V is a measure of input, and U is a measure of output.

Figure 1 The comprehensive meaning and definition of perceived value.

Figure 2 Analysis of consumer information processing process: input, attention, understanding, participation, memory.

Figure 3 Consumer purchasing patterns and negative review impact patterns.

SOR model design

The SOR model is a foundational framework employed for examining consumer behavior patterns. It constitutes a study of how individuals process information from the vantage point of cognitive psychology (Hussain et al., 2023). The SOR model posits that human behavior is a consequence of stimuli arising from the interplay between consumers’ internal psychological processes and external interactions, which subsequently instigate consumer motivation. This motivation, in turn, leads to the formulation of consumer consumption decisions, culminating in shopping behavior. Following the shopping experience, consumers engage in assessments based on their perceptions (Grădinaru et al., 2022; Alanadoly & Salem, 2022). The SOR model is grounded in three key elements: stimuli, organism, and response, which serve as the focal research variables and components (Karim et al., 2021). A detailed explanation of the SOR model structure and its specific meaning is shown in Fig. 4. The SOR model structure and model variable definitions based on negative evaluation are shown in Fig. 5.

Figure 4 SOR model analysis: Detailed explanation of the structure of stimulus, body, response, and its specific meaning.

Figure 5 SOR model based on negative evaluation: Detailed explanation of stimulus, organism, response, and their definitions.

The SOR model, when applied to negative evaluations, is formally established. In order to investigate the relationship between negative evaluations, consumer perception of risk, and consumer behavior, statistical analyses, including the Pearson correlation coefficient (PCC) and factor analyses, are conducted.

Suppose there are two variables, n and m.Cov is the covariance, and Q is the standard deviation. Then, the PCC Pn,m is calculated as shown in Eq. (2). (2) Pn,m=Covn,mQnQm.

When two variables are independent of each other, Pn,m = 0. However, Pn,m = 0 does not confirm that the two are independent. When Pn,m < 0, there is a negative correlation between the two. When Pn,m > 0, there is a positive correlation between the two.

Factor analysis is a statistical method to study the common factors of variables. Suppose there are x samples, and the sample dimension is unified to y. When x is small, large likelihood estimation is used to estimate the mean U and variance.

(3) U=1x∑i=1xni

(4) E=1x∑i=1xni−U.

In Eqs. (3)–(4), E is a covariance matrix, and E is irreversible.

LR model design

The main reason for choosing a logistic regression model is to explain or predict specific types of consumer behavior, thereby gaining a deeper understanding of the events that occur in the online shopping environment. In addition to focusing on consumer decisions to buy or not, this study emphasizes the potential correlation between these behaviors and the online shopping network’s security. It captures and quantifies the influence of various factors on consumer behavior, and subsequently deduces the impact of these behaviors on the security of online shopping networks through the logistic regression model.

The LR model represents a nonlinear regression model extensively employed for the assessment of risk factors, forecasting event occurrence probabilities, and sample classification. The applicable conditions, advantages, and disadvantages of the logistic regression model are shown in Fig. 6.

Figure 6 Analysis of applicable conditions, advantages, and disadvantages of the Logistic regression model.

Based on the above content, this study employs a logistic regression model for a comprehensive consumer behavior analysis to better understand and predict consumer shopping decisions. In the model, after careful consideration, this study mainly focuses on consumer behavior as the dependent variable. The following outlines the process of constructing the model:

Suppose there is a series of factors that affect the result with M, N1, N2, …, Ns as a result. Then, the multivariate linear relationship between the dependent and independent variables is calculated, as shown in Eq. (5). (5) M=ϑ0+ϑ1N1+ϑ2N2+…+ϑsNs.

In Eq. (5), ϑ0, ϑ1, ϑ2, …, ϑs are the parameters of the model. When it is translated into a logarithmic equation, the relationship between the dependent and independent variables will be clear. (6) logM=ϑ0+ϑ1N1+ϑ2N2+…+ϑsNs.

If the probability of an event occurring is R and the probability of not occurring is 1 − R, the probability of an event A is shown in Eq. (7):

(7) TA=R1−R

(8) R=expϑ0+ϑ1N1+ϑ2N2+…+ϑsNs1+expϑ0+ϑ1N1+ϑ2N2+…+ϑsNs

(9) 1−R=11+expϑ0+ϑ1N1+ϑ2N2+…+ϑsNs.

The LR model can be expressed as shown in Eq. (10): (10) logR1−R=ϑ0+ϑ1N1+ϑ2N2+…+ϑsNs.

The feature selection aims to ensure that the selected feature set can best explain and predict network security risks while enhancing the model’s interpretability and reliability. Firstly, this study consolidates previous research to identify the key factors affecting online shopping network security, including consumer capacity, motivational, and contextual factors. In order to facilitate practical analysis, the study utilizes a Taobao user shopping behavior dataset from the Tianchi platform, encompassing extensive information related to e-commerce shopping, such as user shopping behavior, product details, transaction specifics, and more. Data is subjected to cleaning and preprocessing during the feature selection process, including handling missing data and removing outliers to ensure data quality and accuracy. Subsequently, statistical analysis is performed using the Statistical Package for the Social Sciences (SPSS) 26.0, calculating the correlations between various factors and online shopping network security. This process culminates in the selection of a feature set, identifying the most influential factors on network security. Based on the results of feature selection, a logistic regression model is established. The regression equation derived under the logistic regression model is presented in Eq. (11): (11) lnR1−R=ϑ0+ϑ1N1+ϑ2N2+…+ϑsNs.

In Eq. (11), R is the probability of online shopping network risk. N1, N2, …, NS are the variables affecting the network security of online shopping. ϑ0 is the constant term. ϑ0, ϑ1, ϑ2, …, ϑs is the regression coefficient. The specific variable settings of the Logistic regression model based on online shopping network security risks are shown in Fig. 7.

Figure 7 Specific variables of the logistic regression model based on factors affecting online shopping network security risk.

In Fig. 7, the factors influencing online shopping network security primarily encompass consumer capability, motivational, and contextual factors. The reasons for selecting these variables are as follows: 1. Weak consumer perception capabilities may lead to consumers not fully recognizing the potential risks of online shopping. If consumers fail to perceive the risks of information leakage adequately, they might not take appropriate preventive measures, such as using strong passwords and secure payment methods, making their information more vulnerable to threats. Additionally, weak consumer perception capabilities could result in consumers overlooking network security factors in their shopping decisions, as they may prioritize price, convenience, or product features. This might lead them to choose less secure shopping channels or engage in transactions with untrustworthy sellers, thereby increasing network security risks. 2. In the pursuit of ultimate speed, consumers may not pay special attention to the choice of logistics companies. If the information systems of the logistics companies chosen by consumers have vulnerabilities, data leakage and other issues may occur, leading to network security risks. 3. The allure of low prices may prompt consumers to seek lower-priced goods and services when shopping. However, this could also lead consumers to shop on cheaper platforms with fewer security measures. In some cases, this could lead to occurrences of online fraud, such as infringement, false advertising, or unfair competition. 4. User reviews are typically related to product quality and consumer satisfaction. However, with the advancement of technology, cases of fake or abused reviews are gradually increasing, which can also affect network security in online shopping. 5. Brands are usually associated with product reputation and market position. Some well-known brands may become targets of cyberattacks, leading to the leakage of consumers’ personal information and posing network security risks. 6. The convenience of the shopping experience has a certain relationship with network security. The smoothness and simplicity of the shopping process can reduce security risks caused by shoppers’ negligence.

Standard error is an indicator used to measure the dispersion of data points from the sample mean in statistical samples. In the logistic regression model, standard error is used to assess the precision and statistical significance of regression coefficients. A smaller standard error indicates more precise estimates of the regression coefficients. The calculation of standard error is shown in Eq. (12): (12) SE=sqrt∑x−x ˇ2n−p.

In Eq. (12), x represents the observed dependent variable value, x ˇ represents the dependent variable value predicted by the Logistic regression model, n represents the sample size, and p represents the number of independent variables in the model.

Experimental data design

The DEA model, the SOR model of consumer perception under negative evaluation, and the Logistic regression model for online shopping network information security based on consumer perception risk all use the Tianchi database. The Tianchi dataset is a scientific research data platform open to the public by Alibaba Group, primarily consisting of three categories: “Tianchi competition dataset”, “Alibaba research paper dataset”, and “Alibaba business dataset”. The Tianchi dataset mainly features ‘industry-scarce’ datasets that cover most of Alibaba’s business areas, including e-commerce, finance, healthcare, industry, transportation, and more. The e-commerce domain contains a dataset of Taobao user shopping behavior, including data from approximately one million user behaviors and a dataset of approximately 30,000 negative comments on Taobao e-commerce. The data selected for this research is from 2017 to 2021 and includes publicly available user behavior and negative comment databases on the Taobao platform, with no involvement of human experiments or participants. It does not collect any data that can identify personal identities and is not associated with any potential limitations or biases related to data sources.

The models used in this study primarily rely on Python 3.6.13 and SPSS 26.0 software. The operating system is Windows 10, 64-bit, with 16GB of installed memory and an Intel i7 processor. Python is a popular high-level programming language known for its simple and readable syntax, cross-platform support, and versatility. It is widely used in web development, data analysis, scientific computing, artificial intelligence, and automation tasks. Python 3.6.13 is a specific version within the Python 3.x series, introducing significant language and library updates. SPSS is a powerful statistical analysis software widely used in social science research, business analysis, and data mining. SPSS provides a wide range of statistical analysis tools, including descriptive statistics, inferential statistics, regression analysis, factor analysis, cluster analysis, and multidimensional data analysis, enabling users to easily process and analyze various data types. It features a user-friendly interface that allows users to perform data visualization and report generation, making it accessible to non-statisticians. SPSS offers powerful tools for data analysis and decision support, serving as an invaluable tool for researchers and analysts.

This study conducted data preprocessing using Python 3.6.13. Initially, relevant data processing and analysis libraries were imported in Python, and the original data was loaded using the panda’s library. Subsequently, data cleansing procedures were applied, which included removing duplicate records using deduplication functions, converting date data into the proper date format using time series transformation functions, rectifying format errors, filling missing data using the fillna function, and addressing missing values through the use of the value function. Following this, outlier detection was performed using Z-scores, and outliers were removed based on Z-score criteria employing boolean indexing. Subsequently, feature selection was carried out in alignment with the research requirements, and data dimensionality was reduced using principal component analysis (PCA) while ensuring the preservation of user privacy through anonymization. Finally, the dataset was partitioned into training and testing sets, with the training dataset comprising 1,052 data points and the testing dataset comprising 263 data points.

SPSS 26.0 was used for statistical analysis of the data in this study. The data was first imported into SPSS, and the variable types, data distribution, and potential outliers were checked. Descriptive statistical analysis was then performed to generate a statistical summary of the dataset, including mean, median, and standard deviation, to understand the data’s characteristics. Finally, a logistic regression model was built using SPSS. The network security analysis of online shopping consumer perception using the logistic regression model and the future research roadmap for network security services are presented in Fig. 8.

Figure 8 Research roadmap for online shopping consumer perception analysis with the introduction of the logistic regression model and future network security service technologies.

Analysis of online shopping consumer perception and network security results under negative reviews

Analysis of SOR model results based on negative reviews

In this study, the SOR model, specifically tailored to negative reviews, is employed to examine the impact of unfavorable feedback on consumer perception and behavior. The primary research variables of interest are consumer perception and behavior within the context of online shopping. The comprehensive analysis of online shopping consumer perception is conducted through an exploration of negative reviews on the Taobao platform and the associated product’s consumer behavior data spanning the period from 2017 to 2021. Figure 9 reveals the results of the SOR model analysis based on negative reviews.

Figure 9 Analysis results of the SOR model based on negative reviews.

In Fig. 9, the findings illustrate a discernible correlation between negative reviews and three distinct consumer behaviors. Notably, the quantity of negative reviews significantly influences the tendency to delay consumption, as indicated by a correlation coefficient of 0.41. Meanwhile, the intensity of negative reviews exerts a pronounced impact on the inclination to reject consumption, with a correlation coefficient of 0.38. Furthermore, the length of negative reviews significantly affects the proclivity for opposing consumption, displaying a correlation coefficient of 0.38. In sum, it is evident that the length, intensity, and quantity of negative reviews collectively manifest a detrimental effect on consumption, thereby diminishing consumer purchase intent. The correlation analysis results of PR reveal associations between PR, consumer behavior, and negative reviews. Specifically, PR exhibits a strong correlation between delayed shopping behavior and the number of negative reviews, featuring correlation coefficients of 0.41 and 0.4, respectively. Conversely, its impact on opposing consumption behavior is relatively modest, with a correlation coefficient of only 0.27. Overall, it is evident that negative reviews contribute positively to the perception of consumer risk. These findings underscore the presence of a causal relationship between PR and negative consumer behavior.

Results analysis of LR model for NS

This study has devised a LR model with the primary aim of investigating the network security aspects of online shopping. It centers its focus on network security as the principal research variable while considering consumers’ individual characteristics, aptitude factors, motivational elements, and conditional factors as independent variables. The in-depth analysis of online shopping NS, contingent upon consumer perception risk, is conducted by scrutinizing negative reviews on Taobao platforms and the associated product consumer behavior data spanning the years 2017 to 2021. Figure 10 plots the analysis results of the online shopping NS-LR model based on consumer perception risk.

Figure 10 Analysis results of the LR model.

In Fig. 10, the convenience of consumer purchases has a relatively minor impact on online shopping security. The relationship between the two is not particularly significant. Enhancing or reducing the convenience of consumer purchases does not yield consequential effects on online shopping security. However, delivery speed exhibits a notable association with online shopping security, which may be attributed to the considerable proportion of food items in online shopping. The regression coefficient for buyer reviews stands at 0.12, with a standard error of 0.135 and an approximate p-value of 0.05. This data indicates a significant relationship between buyer reviews and online shopping security. Enhancing the authenticity of reviews may promote online shopping security and reduce associated risks. Brand and price demonstrate a significant relationship with online shopping security, but both variables exhibit regression coefficients less than zero. The data suggests that these factors do not completely guarantee online shopping security but decrease the probability of online shopping risks. There is a substantial relationship between consumer perception and online shopping security. Strengthening consumer perception can enhance the capacity to process risk-related information, subsequently bolstering resistance to online shopping risks and elevating online shopping security. Based on the results of statistical significance, it is evident that delivery speed, reviews, brand, price, and consumer perception all exert a significant influence on online shopping security, with consumer perception having the most substantial impact. This underscores the importance of enhancing consumer perception to effectively heighten online shopping security. However, the impact of purchase convenience on online shopping security lacks statistical significance. This discovery offers valuable insights into the factors influencing online shopping security and contributes to the ongoing efforts to enhance and maintain online shopping safety. In summary, online shopping security is influenced by a multitude of factors, and among them, consumer perception holds considerable sway compared to other variables.

Correlation analysis

In order to further explore the relationships between various factors influencing online shopping security, the PCC is applied to conduct a correlation analysis between brand, price, delivery speed, and online shopping security. This coefficient is employed to measure the strength and direction of linear relationships between variables. Table 1 presents the experimental results.

Table 1 Correlation analysis among factors affecting online shopping security.

Variable names	Correlation coefficients	Standard error	P value	
Brand	0.25	0.08	0.02	
Price	−0.18	0.06	0.08	
Delivery speed	0.30	0.09	0.01	

Table 1 indicates that there is a positive correlation between brand and online shopping security, with a correlation coefficient of 0.25. This suggests that an enhancement in brand reputation may positively impact online shopping security. There is a negative correlation between price and online shopping security, with a correlation coefficient of −0.18, indicating that a price reduction may decrease the potential risks associated with online shopping. Delivery speed shows a positive correlation with online shopping security, with a correlation coefficient of 0.30, implying that faster delivery speeds may contribute to an improvement in the security of online shopping.

Discussion

As internet technology continues to integrate into various sectors, e-commerce plays an increasingly crucial role in economic development. Online shopping has gradually become a prevalent business model. However, with the expansion of online shopping, consumers are exposed to a range of potential risks that may have adverse effects on individuals and hinder the growth of the online shopping industry. Therefore, a thorough investigation into online shopping security is deemed necessary. This study employs the DEA model to validate the interaction between evaluation and stimulation and constructs the classical SOR model. Negative online shopping reviews, consumer perceived risks, and consumer behavior are taken as research variables, and attribution theory is applied for regression analysis to explore the relationships among these three variables. Consumer perceived risk acts as the mediating variable, negative reviews as the independent variable, and consumer behavior as the dependent variable. Ultimately, the LR model is used for empirical analysis to identify factors that may influence online shopping security. Unlike previous studies, this study integrates negative reviews and consumer perceived risks, and analyzes the impact of negative reviews on consumer behavior in the online shopping domain. Factors such as negative reviews’ length, quantity, and intensity are examined. The results highlight that negative reviews tend to decrease consumers’ inclination to engage in online shopping, consistent with the concept of consumer perceived risks. The study establishes an online shopping security model, confirming the associations among consumer perceived risks, negative reviews, and consumer shopping intentions. In terms of consumer perception, it opens a new avenue for enhancing online shopping security and mitigates risks associated with online commerce. The work emphasizes that enhancing online shopping security can be achieved by strengthening consumer perception. Collecting negative review data increases consumers’ perception of the risks associated with purchasing high-risk products, effectively reducing consumer enthusiasm for purchases. Additionally, maintaining regulations on product evaluations ensures the authenticity of product information, thereby enhancing the security of online shopping and consumption, serving as a constructive measure to uphold online shopping security.

Conclusions

Against the backdrop of continuous development in the economy and internet technology, the significance of online shopping in people’s lives has become increasingly prominent. In order to enhance the security of online shopping and ensure the safety of consumers in the online business domain, this study combines the DEA and SOR models with a logical regression model. The purpose is to explore the relationships among consumer perception, online shopping security, and consumer shopping behavior. The results reveal that the length, intensity, and quantity of negative reviews contribute to reducing consumer purchase intentions, thereby adversely affecting consumer behavior. This implies that when faced with negative reviews, consumers become more cautious, adopting a skeptical attitude towards their shopping activities, consequently decreasing their inclination to make purchases. Consumer perceived risk plays a crucial mediating role between negative reviews and consumer behavior. Negative reviews positively influence consumer perceived risk, and a causal relationship exists between perceived risk and consumer negative behavior. This suggests that enhancing the ability to perceive risk in consumers can inhibit unfavorable shopping behavior and improve shopping security.

From the perspective of online shopping security, factors such as delivery speed, shopper reviews, brand reputation, price, and consumer perception are closely related to online business security. Among these factors, consumer perceived risk has the most significant impact on online shopping security. Therefore, safer online shopping behavior can be promoted by strengthening the ability to perceive risk. This includes increasing consumer awareness of online security, ensuring consumers understand the potential risks of online shopping, identifying risk issues, and avoiding the use of hazardous online software, tools, or technologies. In this way, potential risks of online security can be alleviated, ensuring the safety of online shopping. It is important to note that this study has some limitations. The measurement of negative reviews is still somewhat inadequate, as the analysis is limited to three specific dimensions. The length and intensity of evaluations may be subject to subjective influences, and the dataset used in the empirical research on consumer perception and online shopping security is relatively limited, with room for improvement in variable selection.

Supplemental Information

Supplemental Information 1 Raw data of the images

The relevant data of the images in this article can be opened and viewed using VISIO and ORIGIN software.

Click here for additional data file.

Supplemental Information 2 DEA model code

The paper employs a Data Envelopment Analysis (DEA) model to confirm the relationship between evaluation and stimuli.

Click here for additional data file.

Supplemental Information 3 Logistic regression model code

A Logistic regression model is employed to empirically analyze factors that may influence online shopping security.

Click here for additional data file.

Supplemental Information 4 SOR model code

A Stimuli-Organism-Response (SOR) model is established, incorporating variables such as negative online shopping reviews, perceived consumption risk, and consumer behavior.

Click here for additional data file.

Additional Information and Declarations

Competing Interests

Author Contributions

Data Availability

The authors declare there are no competing interests.

Feng Lu conceived and designed the experiments, performed the experiments, analyzed the data, performed the computation work, prepared figures and/or tables, authored or reviewed drafts of the article, and approved the final draft.

The following information was supplied regarding data availability:

The data is available at figshare: Lu, Feng (2023). data.zip. figshare. Dataset. https://doi.org/10.6084/m9.figshare.24098112.v1.

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
