# Peer review of "Online shopping consumer perception analysis and future network security service technology using logistic regression model"

_PeerJ Computer Science, doi:10.7717/peerj-cs.1777_

## Round 0.1 · original submission · Major Revisions

Dear author,

After meticulous consideration by the team of reviewers, it is suggested that you pay attention to these and provide a significant improvement to the document. A set of very pertinent issues are highlighted that deserve attention.

**Language Note:** The review process has identified that the English language must be improved. PeerJ can provide language editing services - please contact us at [email protected] for pricing (be sure to provide your manuscript number and title). Alternatively, you should make your own arrangements to improve the language quality and provide details in your response letter. – PeerJ Staff

Reviewer 1 ·

Basic reporting

1. The abstract needs to be precise.
2. The outcome should be mentioned in the abstract in percentage.
3. The citation style in the text is not correct. Need to keep one style in the entire paper.
4. The contribution of the authors or novelty of the work is not mentioned. Need to include a subsection after the introduction as “Our contribution” and need to mention it in clear bullet points.
5. The organisation of the papers is missing.
6. The literature review is very short.
7. The research gap is not identified. The literature must have a clear structure of the existing work. E.g., the methodology, objectives, performance measures and most importantly the limitations of the existing schemes.
8. Haven’t seen any scheme for the year 2023 in the literature review part.
9. The authors need to include feature selection, and how they calculate the sum square error.
10. The resolution of the figures is too low and unreadable.
11. The conclusion needs to be revised and precise.
12. Future research work should be included.
13. References need to be updated.

Experimental design

Defined but not clear to understand.

Validity of the findings

N/A

Reviewer 2 ·

Basic reporting

First and foremost, the paper's purpose and relevance aren't very clear. It attempts to connect online shopping consumer perception and network security, but the connection is quite muddy. The introduction lacks a compelling narrative, leaving readers puzzled about the paper's objectives. Also, both the title and abstract should be more accurate and descriptive.

Another significant problem is the clarity and formatting of the diagrams. They're quite unclear and don't adhere to formatting guidelines. These need to be revised to enhance their clarity and integration within the text. Captions should also be more informative.

The explanation of the logistic regression model used in the study is not sufficiently detailed. Readers need a clearer understanding of the model's setup, its variables, and why they were chosen.

To add to these issues, the paper's overall organization could be improved, and language and grammar need further proofreading. Also, the references should be more comprehensive and relevant to the topic at hand.

In summary, this paper requires substantial revisions before it can be considered for publication. I recommend that the authors clarify the paper's focus, enhance diagram clarity and formatting, provide a detailed explanation of the logistic regression model, ensure relevance to the chosen topic, and improve overall organization and language. I look forward to seeing an improved version after these issues are addressed.

If you need any further clarification, please don't hesitate to reach out to me.

Experimental design

The experimental section of the paper titled "Online Shopping Consumer Perception Analysis and Future Network Security Service Technology Using Logistic Regression Model" has several critical shortcomings that require attention and clarification.

First and foremost, the experimental design lacks clarity regarding the specific features used in the logistic regression analysis. It is essential to provide a clear description of the chosen features, their relevance to the research question, and how they were selected.

Additionally, the paper does not adequately explain any dimensionality reduction techniques that were employed, which is crucial for understanding how the data was prepared and processed. A detailed explanation of any feature selection or dimensionality reduction methods is necessary to provide transparency and reproducibility of the results.

The paper also falls short in providing information about the platform or software used to execute the logistic regression. Readers should be informed about the software environment, version, and relevant libraries or tools to facilitate replicability and understanding.

Furthermore, the implementation of the dataset is unclear. Readers should have a clear understanding of how the dataset was collected, pre-processed, and made ready for analysis. This includes details on data cleaning, transformation, and any missing data handling procedures.

Finally, the paper lacks a structural diagram that could help readers visualize the entire experimental process. A diagram or flowchart illustrating the data pipeline, feature selection, dimensionality reduction, logistic regression modeling, and any other relevant steps would greatly enhance the paper's clarity and transparency.

In conclusion, the experimental section of the paper requires significant improvements in terms of explaining the chosen features, dimensionality reduction techniques, platform used for logistic regression, dataset implementation, and the inclusion of a structural diagram. Addressing these issues will enhance the paper's quality and the comprehensibility of the experimental process.

Validity of the findings

The validity of the findings in the paper raises some concerns as the section does not sufficiently support the results presented. To establish the credibility of the findings, it is crucial to provide a comprehensive discussion on the methodology, data collection, and analysis. However, the paper falls short in this regard. It lacks transparency in explaining the data sources, sample size, and any potential biases or limitations in the data. Moreover, there is an absence of statistical tests and measures of significance to demonstrate the robustness of the results. The paper should address these issues by providing a more detailed account of the research methods, data sources, and the statistical analyses performed. A clear and transparent presentation of these elements is essential for establishing the validity and reliability of the study's findings.

Additional comments

Data Source Clarity: The paper would benefit from a more explicit description of the data sources used. It is essential to detail where the data for online shopping consumer perception and network security service technology was collected, how it was obtained, and any potential limitations or biases associated with the data sources.

Statistical Significance: The paper should incorporate statistical tests and measures of significance to support the findings. This is essential for ensuring that the results are not just due to chance and have practical relevance.

Integration of Dimensionality Reduction: If dimensionality reduction techniques were applied, the paper should provide a thorough explanation of the specific methods used and the rationale behind their selection. This is crucial for understanding how the data was transformed and prepared for analysis.

---

## Round 0.2 · Minor Revisions

Dear authors, you are advised to critically respond to all comments point by point, while preparing for the response.

Kind regards

Reviewer 1 ·

Basic reporting

N/A

Experimental design

N/A

Validity of the findings

N/A

Reviewer 2 ·

Basic reporting

The author claims that the problem is the multifaceted nature of identifying consumer perception, where multiple factors exert an influence. Consumers do not hinge their judgments solely on a singular factor during consumption. However, in the proposed model, there is only a dependent variable called consumer behavior which is being handled by applying logistic regression. It is not clearly stated how it is enhancing the online shopping network security as claimed by the author.
The scope of actual extermination seems less as compared to the things that are claimed in the paper.


There are many important models that are handling sequence problems such as LSTM that can better handle the features such as negative comments lengths etc so there is a need to add some information in the Literature review section related to Logistic regression that how they are better to adopt as compared to other sequence-based models.

Experimental design

SPSS and Python both are integrated for the logistic regression model, Data pre-processing steps are very well explained. and data pre-processing and model application are well applied.

Validity of the findings

The paper claims that Brand and price demonstrate a significant relationship with online shopping security, but the validity of the idea is not well related.
The paper also claims that delivery speed exhibits a notable association with online shopping security.
The identified features are not strongly proven to logically enhance the online shopping security of users.
It is lacking with the proof of concept.

---

## Round 0.3 · accepted · Accept

The author made the required changes indicated by the review team, and is therefore in a position to be accepted for publication.